# Effects of a 24-Week Low-Cost Multicomponent Exercise Program on Health-Related Functional Fitness in the Community-Dwelling Aged and Older Adults

**DOI:** 10.3390/medicina59020371

**Published:** 2023-02-15

**Authors:** Filipe Rodrigues, Miguel Jacinto, Nuno Figueiredo, António Miguel Monteiro, Pedro Forte

**Affiliations:** 1ESECS—Polytechnic of Leiria, 2411-901 Leiria, Portugal; 2Life Quality Research Center, 2040-413 Leiria, Portugal; 3Department of Sport Sciences, Polytechnic of Bragança, 5300-253 Bragança, Portugal; 4Research Centre in Sports Sciences, Health, and Human Development, 6201-001 Covilhã, Portugal; 5CI-ISCE ISCE Douro, Higher Institute of Educational Sciences of the Douro, 4560-708 Penafiel, Portugal

**Keywords:** exercise, quasi-experimental, health, physical fitness

## Abstract

*Background and Objectives*: This study aimed to examine the effects of a low-cost multicomponent exercise program on health-related functional fitness in the community-dwelling aged and older adults. As a second objective, this study compared the exercise program between aged adults (<65 years) and those considered elderly (≥65 years). *Materials and Methods*: Forty-eight participants were included in the exercise program, and their mean age was 64.73 years (±5.93 years). The Senior Fitness Tests were performed by each participant. A dynamometer was used to assess hand grip strength, and body composition was assessed considering the body mass index. Paired-sample t test was used to compare data at baseline and after the exercise program, considering the total sample. Afterwards, a 2 × 2 analysis of variance was used to examine differences within and between groups. *Results:* Statistically significant improvements in the chair stand (t = −14.06; *p* < 0.001; d = 0.42), arm curl (t = −12.10; *p* < 0.001; d = 0.58), 2 min step test (t = −9.41; *p* < 0.001; d = 0.24), timed up and go test (t = 5.60; *p* < 0.001; d = 0.19), and hand grip strength (t = −3.33; *p* < 0.001; d = 0.15) were observed. There were also significant differences in the back scratch (t = −6.68; *p* < 0.001; d = 0.18) and chair sit and reach test (t = 5.04; *p* < 0.001; d = 0.05), as well as body mass index (*p* < 0.05). No significant differences were found between groups (*p* > 0.05). *Conclusion*: This study provides evidence that a 24-week low-cost community-based exercise program can improve functional fitness in aged and in older adults. The exercise program supplied the necessary data to construct further randomized controlled trials that can be performed in the community in an environmentally sustainable fashion and applied, not only to the elderly, but also to those transitioning to this age group.

## 1. Introduction

An aging population is no longer an issue of the future, but of the present. Given the United Nations’ prediction that the elderly population would reach 2 billion by 2050, global aging is bringing new realities and challenges to most countries’ economic, social, and health systems [1]. The increasing numbers of adults in the 65-year-old category could also be attributed to developments in medical care and technologies that aid in a variety of everyday activities, hence lessening daily physical demands. The globe has seen massive spending in physical therapy, although prevention may prove to be the most cost-effective strategy. For example, among people aged 80 and up, impairment was more predictive of death than multimorbidity [2]. Survival curves show that more efforts should be made to improve quality of life, as preventing disability is preferable to treating noncommunicable diseases [3]. The elderly wish to live longer and improve or maintain their quality of life as they age. In addition, the majority of them prefer to live at home rather than in nursing homes or daycare facilities [4].

Physical fitness level falls at a consistent rate as people age. Sarcopenia, defined as the loss of muscular strength and mass as a result of biological aging, is one of the leading causes of functional decline and loss of independence in older people, as well as a cause of the increase in the risk of fall and fall-related injuries [5]. The maintenance of flexibility, force and power production, and coordination is dependent on physical activity levels during previous stages in life, such as adulthood [6]. The ability of a joint or group of joints to move through an unlimited, pain-free range of motion is referred to as flexibility. Although flexibility varies greatly amongst people, some minimum ranges are required to sustain joint and total body health. As aging progresses, individuals tend to lose range of motion, decreasing their capacity to perform daily activities [5]. Cardiorespiratory fitness is defined as a component of physiologic fitness that relates to the ability of the circulatory and respiratory systems to supply oxygen throughout the body. Aged and older adults tend to have diminished aerobic function, and this is strictly correlated with physical disability, gait speed and velocity reduction, poor quality of life, and death [7]. Aging is also characterized by multiple transformations in the human body, including changes in body composition, such as increased body fat, which is related to the increase in the incidence of chronic diseases, such as diabetes, chronic lung disease, cancer, and stroke [8]. As a result, it is critical to develop effective interventions that promote physical activity, focusing on health-related components, in order to minimize the risk of fall and injury, as well as chronic diseases, morbidity, and premature mortality [5,6].

Community programs that provide exercise interventions may be helpful in allowing the elderly to live independently with enough physical and functional fitness [9]. These programs are frequently offered primarily for research reasons, with pre-determined inclusion requirements, and they stop when data collection is complete. While this conclusion is fine for research on the effects of exercise on specific groups, it limits the inclusiveness of exercise, which has been marginalized globally. On other occasions, exercise programs are structured to rely on high-cost resistance and cardiorespiratory machines (i.e., treadmills), which is not ecologically feasible, as funds are limited. Community exercise programs supervised by exercise physiologists for aged adults and older, using low-cost equipment like that used in their daily activities (e.g., dumbbells, which are similar to small packages) could be an opportunity to promote functional fitness [10]. Thus, the search for reliable long-term and low-cost exercise interventions is warranted for the purpose of increasing physical and functional fitness among those aged adults intending to be physically active [11].

Exercise in aging adults can reduce functional decline and it is also related with higher autonomy and independence, as well as a decreased healthcare load [12]. Its effects reach beyond functional capacity, with significant implications for lowering fall risk and injuries [13], lowering clinical depression [14], and improving sleep quality [15] in aged adults. Multicomponent exercise programs have been used to improve the functional fitness of the community-dwelling elderly [5,16] because they allow the combination of different exercise regimes in the same exercise routine. This type of exercise training combines different types of exercise (e.g., strength, balance, gait, aerobic, and flexibility) in the same exercise session, and it is a public health program option aiming to maintain or even improve the physical and cognitive functions of community-dwelling aged and older adults [17]. Researchers have described the unique effects of an acute multicomponent exercise program to slow functional decline in older individuals [11,18,19]. This type of training has been regarded as a substantial contributor to improved functional fitness, metabolic results, and cognitive performance [20]. Thus, by considerably improving physiological and psychological results, it is expected that the quality of life will improve as well [21], with the goal of increasing or maintaining independent living and autonomy [5,22]. However, the majority of documented interventions include resistance training and/or cardiorespiratory training with expensive equipment that requires expenditures that are not always available for community objectives [9,23]. Other limitations include the small sample size, the investigation of exclusively older adults, neglecting those individuals transitioning from adulthood to elderly age, and the lack of a pattern in the distribution of the exercise regimes, given that some studies prioritized resistance training [9], while others prioritized gait exercises [17]. In addition, the objective of community exercise programs should prioritize the continuation of the exercise after the completion of the research for community-dwelling aged and older adults to have an opportunity to increase health-related fitness.

To the best of our knowledge, multicomponent exercise programs have been utilized in community-dwelling adults over the age of 65, but protocols are not always described in detail for replication [20,24,25]. Hence, proposed exercise protocols are not always reproducible. This could be associated with the fact that authors may assume that other researchers are familiar with the general exercise protocols and only include essential details for the specific study. Additionally, various studies have tested exercise intervention exclusively in the elderly, excluding those that are transitioning to this age phase [9,10]. This gap should be tested as a means to provide the validity of the same exercise program for both older adults and those approaching this age group. It is critical that community programs involve people of all ages (remembering that physical demands are not limited to those over 65) and that these community exercise programs be adaptable and effective for everyone. Researchers [12,18] of active aging argue that it begins long before people are “chronologically old.” Because exercise improves long-term quality of life [22], this type of physical activity may assist stakeholders in providing efficient interventions to delay or perhaps prevent age-related functional decline in their communities, hence increasing health-related quality of life in community-living older individuals. Furthermore, community fitness programs with an inclusive approach may encourage more older people to engage in regular exercise on the long-term, as autonomous motivation tends to increase with frequent exercise [26].

The present study’s objective was to evaluate if a multicomponent exercise program could increase physical fitness in community-dwelling aged and older adults. Low-cost valid and reliable measures were used to assess participants’ functional fitness and associated measures of quality of life, such as hand grip strength. Furthermore, as a second objective, this study also compared the exercise program between aged adults (<65 years) and those considered elderly (≥65 years). The exercise intervention was anticipated to have a positive and significant effect on physical fitness in community-dwelling older individuals. Despite evidence of physical decline with age, there are no data to determine if the beneficial effects are equal between those in the elderly group and those approaching this age group. Therefore, the current study was conducted to assess how a multicomponent exercise program affected functional fitness in aged populations after 24 weeks of intervention.

## 2. Materials and Methods

### 2.1. Design

This was a 24-week longitudinal study of a multicomponent exercise program in aged adults, including the before and after intervention analyses. All of the evaluations were assessed by two exercise physiologists with research experience, and the data were analyzed blindly by one researcher. The first two and the last two weeks were used for functional fitness assessments. The community program called “+idade + saúde” was carried out in a gym facility at the School of Education of Bragança. The research project received approval by the Scientific Board of the Higher Institute of Educational Sciences of the Douro (nº = 2.576), Research Centre in Sports Sciences, Health, and Human Development (nº UID04045/2020), and adhered to the principles of the Helsinki Declaration [27]. All participants signed written informed consent forms. There was no monetary compensation.

### 2.2. Recruitment

The G*Power 3.1 was used to calculate the required sample size, considering the following parameters: anticipated effect size of f = 0.2, α = 0.05, and statistical power = 0.95; number of groups = 2; number of measurements = 2; correlation among repeated measure = 0.75; nonsphericity correction = 1. The calculations suggested a minimum of 44 participants for the results to be valid and reliable. The calculation of the effect was based on studies using similar protocols and designs converting reported Cohen’s d effect sized into f [28,29,30]. Thus, minimum sample size requirements were respected.

Recruiting the study population is a particular problem in aged population research, given the present means of communication and social isolation. As a result, the exercise program was publicized through regional journals, social media platforms, and flyers placed in traditional shops (e.g., coffee shops, bakeries, hair salons). Individual invitations to engage in the exercise program were also made over the telephone, considering data from the city council of Bragança. The potential participants were informed about the study’s objectives, as well as about the voluntary contribution of participation and the potential risks of physical harm. An inclusive approach was used to recruit as many potential participants as possible from the community. The exercise program group included all participants that met the inclusion criteria and volunteered to participate in this study. The following inclusion criteria had to be observed for objective and safety reasons: minimum age of 50 years, the capacity to stand and walk with or without assistive equipment, not being engaged in any type of exercise program, and living in the community. A history of chronic neuromuscular, cardiovascular, or metabolic illnesses that could pose a danger or a safety risk during classes and/or evaluation periods were considered, and those who did not meet these criteria were excluded from participation for safety reasons. Participants needed to be available to participate in every triweekly session of the physical exercise program, as well as in the evaluation periods. Exclusion criteria were defined as participating in less than 75% of the sessions and/or absence from more than 10 consecutive sessions. The participants were advised to maintain their daily routines regarding physical activity (e.g., gardening, household activities). At baseline and after the intervention, assessments were performed by two qualified exercise physiologists with the support of one senior researcher. The interclass correlation was used to establish intra-observer validity. The correlation coefficient was acceptable (ICC = 0.83). All coexisting diseases or problems associated with the intervention were handled according to standard medical practice and documented as adverse events.

### 2.3. Intervention

The Frequency, Intensity, Type, and Time (FITT) principles were followed in the exercise program according to the American College of Sports Medicine [11]. The exercise intervention was scheduled for three morning sessions per week, including 45–60 min of resistance, cardiorespiratory, balance, agility, and flexibility training. Two experienced exercise physiologists with extensive training in the exercise prescription for adults and elderly individuals observed all sessions and provided helpful guidance and encouragement.

Exercise sessions were held on weekday mornings in a day-off-day sequence, according to the preferences of the participants. Due to the large number of participants, two groups for each age group (i.e., aged adults and older adults) were formed for safety purposes. Each individual in the group performed the same exercises and adaptations were made when necessary. The talk test and the 10-point Borg Perceived Exertion Scale, which are valid, reliable, practical, and low-cost methods for monitoring exercise intensity [31,32], were used to measure the exercise intensity. Looking at the Borg Perceived Exertion Scale, for each item we provided statements for anchoring perceived effort. For example, for scoring 1 we described “hardly any exertion” and for scoring 10 we described “feels almost impossible to keep going”. The exercise physiologist administered the scales to each participant immediately after each component and at the end of the session throughout training sessions. Three distinct sessions were developed and implemented in order to offer participants with a variety of stimuli.

The training sessions lasted 45–60 min and included the following activities: (a) 5–8 min of warm-up, with slow walking combined with light-intensity dynamic stretching exercises and dual-task activities. (b) Activities for cardiorespiratory fitness lasting 15–20 min were created. Walking, jogging, aerodance, or dance exercises were applied for cardiorespiratory stimuli. As the name indicates, walking involved moving forward by taking steps on tippy toes, on heels, with a knee-raising gait around a predefined circuit. Similar to walking, jogging was performed around a predefined circuit. However, we asked participants to increase hip and knee flexion with increased pace. Aerodance consisted in rhythm-based exercises to the beat of the music. Similarly, dance exercises were performed in pairs according to music tempo. The participants had the opportunity to choose music preferences. A set of two workouts lasting at least 8–10 min each was chosen. Specifically, in one session, participants completed the walking and aerodance activity. In another session, participants completed jogging and dance exercises. Lastly, in the third session, participants performed walking and dance exercises. The intensity of the cardiorespiratory training sessions ranged between 6 and 7 (moderate intensity) on the perceived exertion scale and progressed to moderate-to-vigorous intensity (scores of 7 and 8 on the Borg scale) after 12 weeks. (c) Resistance exercise lasted between 15 and 30 min. Bodyweight, ankle weights, rubber bands, and dumbbell equipment were used for resistance exercises. In a circuit, the participants performed one to three sets of resistance exercises. The rest times between sets in the circuit ranged from 40 to 60 s. The chosen exercises targeted the key muscle groups, including the knee flexors/extensors, shoulder abductors/adductors, elbow flexors/extensors, pectoral, and back muscles. Each session comprised four different exercises targeted at different key muscles. Specifically, in one session, participants completed chair squats, seated arm abduction and adduction, arm curls, and shoulder shrugs using dumbbells. In another session, participants performed seated single-leg extension and flexion with ankle weights, arm flexion and extension with dumbbells, and peck deck with rubber bands. On the third session, participants performed standing calve raises, arm curls with shoulder press, and seated rows using dumbbells. The training intensity was increased from light-to-moderate intensity (5–6 points at the Borg Scale) to moderate intensity (6–7 points at the Borg Scale) after eight weeks from the start of the exercise program to allow for optimal adaptation and workout execution. The participants began with a single set of 8 repetitions and subsequently proceeded to three sets of 12–15 repetitions. All participants had the ability to count. If the participant was able to perform 12 repetitions with light-intensity effort, the exercise physiologist would indicate to perform 3 more repetitions. (d) For 5–8 min, static and dynamic balance training was performed using wooden sticks, softballs, and balloons. Throwing and/or catching softballs, as well as single-leg static and dynamic activities with bats and balloons, were used for balance training. The exercise physiologists ensured a 2 m distance between participants for safety precautions to execute safely each balance and agility exercise. (e) At the end of each session, there was a 5 min cool-down phase that included breathing and stretching exercises. Participants repeated each stretch 3–4 times. When executing a static stretch, the muscle was extended across the joint, kept in a position of low-to-mild discomfort for 15–20 s, and then released. The resting duration between stretches ranged for 15–20 s. Thus, flexibility exercises were done at a 1:1 (active:relaxed) ratio and focused on low-intensity stretching on recruited muscles during the previously executed exercises.

The exercise intervention was created to meet the ACSM [11] and American Heart Association [33] exercise and physical activity guidelines for aged people [34]. Every training attendance was recorded. The exercise physiologists documented adherence to the exercise intervention program in a weekly register. Persistent participants were those who attended at least 75% of all training sessions.

### 2.4. Outcome Measures

All measures were performed by the same evaluator on two separate occasions: the first one prior to the start of the intervention and the final one after 24 weeks of physical exercise. The assessments were carried out in a circuit designed to reduce fatigue effects, and the identical circumstances were maintained for each test during the testing time. On test day, subjects underwent an 8–10 min warm-up conducted by one of the exercise physiologists before proceeding to complete all outcome measures.

The Senior Fitness Tests [35], created and tested to evaluate several health-related physical fitness components, were used. These tests are non-invasive and are reliable and valid to determine the functional fitness of the elderly and aged individuals. Lower limb strength, upper body strength, lower body flexibility, upper body flexibility, agility, balance, and cardiorespiratory fitness were evaluated. Participants received instructions and demonstrations as recommended by Monteiro et al. [16]. The 30-s chair stand test was used to assess lower body strength. Participants were instructed to sit on a 43 cm high chair with their arms crossed and against their chest. Then, they executed as many repetitions (“stand ups”) as possible in 30 s. The arm curl test was used to assess upper body strength. Participants used a 2 kg (for women) and a 3.5 kg (for men) dumbbell to do as many biceps curls as they could in 30 s. The chair sit and reach test was used to examine lower body flexibility. The best distance attained between the outstretched fingers and the tip of the toe, measured to the closest 0.5 cm, was used to calculate the distance. The back scratch test was used to examine upper body flexibility. The score was determined by measuring the shortest distance between the outstretched middle fingers to the closest 0.5 cm. The timed up and go test was used to assess agility and balance. The time it took to rise from a seated posture, walk 2.44 m, turn, and return to a seated position was measured to the closest 1/10th of a second. The 2 min step test was used to assess cardiorespiratory fitness. Participants were asked to march in place for two minutes, lifting the knees to the height of the mark on the wall. Resting was allowed by holding onto the wall. For safety and physical fitness capacity, the 2 min step test was performed on a different day from the remaining Senior Fitness Tests, and it was conducted only twice. The best score was considered. Following a demonstration by the researcher, the 30 s chair stand and arm curl involve a practice trial of two measures, followed by one test trial. In the chair sit and reach, back scratch, and timed up and go tests, the score from the best attempt was used to measure physical fitness components. Rikli and Jones [35] provide comprehensive information on test administration and methods.

The digital scale Tanita BC-545 (Tanita, IL, USA) was used to determine weight (BMI). Participants were requested to have their usual breakfast at 8:00 a.m., at least two hours before measurement (10:00 am). The stadiometer SECA 216 (SECA, Hamburg, Germany) was used to determine height. The BMI was calculated using the standard formula: weight (kg)/height^2^ (m).

Hand grip strength was determined using a certified dynamometer (CAMRY EH101, Guangdong, China) validated for several populations, including adults and older adults. Two measurements were conducted using the dominant hand and the best score was considered for assessment. Hand grip strength was assessed in a sitting position, with the shoulder in a neutral position, the elbow extended at 180°, and the forearm and wrist in neutral positions.

Attendance rates for both groups were calculated by dividing the number of exercise sessions performed by each participant by the full number of sessions they were expected to perform throughout the study (3 sessions per week × 24 weeks = 72 sessions). The possible number of falls, visits to the emergency, hospitalization, and length of stay were also noted.

### 2.5. Statistical Analysis

The IBM SPSS STATISTICS version 27 (Chicago, IL, USA) for Windows was used to analyze all data. Descriptive statistics were reported, and to determine the statistical significance of deviation from normal distribution, the skewness and kurtosis estimates were calculated. Scores below |2.00| suggest a normal distribution [36]. For all tests, the significance level to reject the null hypothesis was set at 5%. To begin, global sample analysis protocols were created. A paired sample repeated measures test was used to examine for differences in the dependent variables (data at baseline and after the 24-week exercise program) across the total sample. Cohen’s d effect size was calculated between time points, and thresholds were set at 0.2, 0.5, and 0.8 for small, medium, or large effects, respectively. Afterwards, a 2 × 2 analysis of variance was conducted to examine differences within and between. For these analyses, Levene’s test was used to examine the assumption of homoscedasticity for the two independent variables with two levels (i.e., group and time). Significant analysis of variance main effects and interactions were followed up by Bonferroni-adjusted post hoc tests to analyze pairwise comparisons. The significance level was set at *p* < 0.05. Partial eta squared effect sizes were computed, with the following assumed reference values: “small” effect = 0.01, “medium” effect = 0.06, and “large” effect = 0.14.

## 3. Results

Forty-eight participants were included in the exercise program (37 female; 11 male), and their mean age was 64.73 years (±5.93 years). While there were no dropouts recorded from the exercise program, 3% of missing completely at random values were identified. The expectation–maximization approach was used to handle missing data. Mean attendance rates were 83%, ranging from 75% to 90%. Regarding the number of falls and fall-related injuries, the exercise physiologists reported no falls during the exercise intervention. Outside of the fitness program, participants reported 2 falls with no fall-related injury which could inhibit continuation of the exercise program. There were no recorded emergencies or hospitalizations. Data at baseline displayed a normal distribution (Table 1). Significant improvements were observed in the chair stand, arm curl, 2 min step test, timed up and go test, and hand grip strength (*p* < 0.001), with small and medium effect sizes ranging from d = 0.05 to d = 0.58, respectively. There were also improvements in the back scratch and chair sit and reach tests, as well as BMI (*p* < 0.001). The arm curl walk test showed the greatest difference between baseline and after intervention (d = 0.58), and the back scratch showed the most significant Δ% (see Table 1).

Results from the descriptive statistics of each group at baseline and after exercise program are reported in Table 2. The <65 years age group (*n* = 22) comprised 16 female and 6 male and the mean age was 59.50 years (SD = 3.89 years). The ≥65 years group (*n* = 26) comprised 21 female and 5 male and the mean age was 69.15 years (SD = 2.98 years).

Results from the repeated measures multivariate of variance analysis (time vs. groups) are reported in Table 3. There were significant changes in all health-related components in both groups in the time condition (*p* < 0.001). However, there were no significant differences between groups and in the time*group setting. Hence, both groups statistically increased health-related components. Effect sizes varied between small and large.

## 4. Discussion

This study aimed at examining the effects of a multicomponent exercise program on health-related functional fitness in community-dwelling aged and older adults. As a second objective, the present study compared the exercise program between aged adults and those considered elderly. The findings show that a multicomponent exercise program involving low-to-moderate intensity resistance, cardiorespiratory, balance, agility, and flexibility training, with an increase to moderate-to-vigorous intensity exercises performed three times per week for 24 weeks, provides a significant benefit and can help reverse the functional decline associated with aging. No significant differences were found between groups. Thus, these findings support the original hypothesis that the exercise intervention has a positive and significant effect on physical fitness in community-dwelling aged and older adults.

Participants in this community exercise program obtained significant gains in multiple functional capacities with a very minimal investment in equipment, when compared to typical strength and/or cardiorespiratory training using high-cost devices [5]. Indeed, the gains were similar to those reported in a recent review of 28 studies on resistance training [37]. As a result, multicomponent exercise programs may be appropriate, as there are indisputable connections between high physical activity and low sedentary behaviors, as well as improved muscle strength in physically active older adults [38]. Multicomponent exercise programs, as documented in recent meta-analyses [39,40,41], are effective in improving numerous elements of health-related physical fitness.

The multicomponent exercise program considerably improved lower and upper limb strength. This could be explained by the stimuli imposed by the exercise program and the compliance with exercise prescription guidelines [34]. As participants performed strength exercises, the demands placed on their muscles stimulated muscle fibers, leading to an increase in muscle strength. Progression training increased across the twenty-four weeks in which participants performed a higher volume of upper limb and lower limb exercise. These findings appear to be consistent with the Monteiro et al. [16] study. In addition, current results support those described in a recent meta-analysis [41] showing the importance of multiple sets in producing benefits in muscle strength. Multicomponent exercise programs that include strength training appear to improve upper and lower limb strength if stimuli are maintained across time [9,42,43]. It is important to note that the exercise intensity of strength training was tailored to the individual’s physical abilities and limitations and that proper instruction and supervision were recommended, since the exercise physiologist controlled exercise intensity.

Participants increased their flexibility, since lower and upper limb tests revealed significant improvements. While there are some inconsistencies regarding the benefits of flexibility training [44], current results support previous research examining the positive effects of exercise on flexibility [9,41]. During the flexibility training, participants performed several exercises that elicited increased range of motion. The significant improvements could also be related to the exercises conducted in the strength component, as range of motion is increased by this sort of training [9]. As muscle fibers tend to increase, there is some plasticity regarding the muscle contraction and stretching [41]. Thus, the inclusion of stretching exercises in a multicomponent exercise program can increase flexibility in aged and older adults.

Significant differences were also found between baseline and after the exercise program in the cardiorespiratory fitness component. These results are in accordance with previous literature suggesting that aerobic training is essential to promote improvement in cardiorespiratory fitness [17,45]. This could be explained by the exercises induced during the program and the compliance of the participants with the stimuli. The intervention included cardiorespiratory activities which are similar to those performed by the aged adults (e.g., walking, jogging) as well as those that are enjoyed by them in recreational activities, such as aerodance or dance. Hence, stimulation of fun activities that are promoted as a means to increase physical fitness [46], specifically those indicated to promote improvements in aerobic capacity, was able to induce enhancements in gait in the participants of this study.

There were significant variations between the baseline and after the exercise program in agility and balance. In can be speculated that the observed significant exercise effect was due to the nature of the induced exercises. Because this program included exercises similar to daily activities, it is likely that participants took advantage of the beneficial effect of agility and balance training. A supervised and safe environment has a high potential for exercise training, since it enhances attention and motivation across participants, which may lead to better performance than exercising alone in a home setting. The present study´s results are in accordance with previous literature [9,47], showing that strength training could also be associated with the increase in agility and balance, since resistance training involves force production movements and explosive moments, such as knee extension in a standing position, which is related to agility and balance.

With growing evidence emphasizing exercise as a strong predictor of positive health outcomes in aged and older adults, researchers have sought to design and test multicomponent interventions to improve functional fitness and to reduce falling risk and fall-related injuries [24,25]. To date, research has demonstrated that short-term exercise programs have advantages, but the long-term implications of these protocols have greater benefits [44,48]. The present intervention was able to enhance physical fitness and to maintain autonomy by combining a multicomponent exercise program with low-cost equipment over a course of twenty-four weeks. There were no falls linked with this intervention, and participants did not report falls on subsequent occasions. There are various possible causes for this intervention’s success. As suggested by other studies, a community-based exercise program with an inclusive approach was provided to older individuals, who may have a higher propensity to engage and to intend to remain in their homes [22,24]. Furthermore, huge group dynamics from exercise sessions may have resulted in positive feedback, as well as reinforcement from the maintenance of high attendance, perhaps increasing commitment to the exercise program.

The current findings can be used to plan and implement future useful public health interventions for community-dwelling older individuals, which could reduce prospective healthcare expenses associated with aging (e.g., falls, hospitalization, medications). Long-term systematic interventions should investigate the success of these programs, particularly in communities with varying features. Not only should exercise interventions be inclusive, but their execution should also be tailored according to the needs of this community [20].

### Study Limitations

In addition to providing valuable information, this study also has some limitations which should be considered. Due to the objective and lack of randomization, this was a quasi-experimental study. Thus, the results should be further explored, as there is no certainty if the improvement in functional fitness would translate into better long-term outcomes, using a randomized controlled approach. In addition, the detraining effect of this exercise program was not evaluated. Future studies should also consider mid-intervention data collection for tracking progress through the training program. More studies are warranted to examine the efficacy of this type of intervention using detraining assessment equal to intervention period. While there was no control group to compare the benefits of the exercise program on functional fitness, studies like the present one give preliminary evidence of how an exercise program can be effective in promoting health-related benefits. Nonetheless, this study has some advantages, notably its uniqueness. The majority of exercise programs in older individuals have been conducted in laboratory settings, limiting their replicability in the community. In addition, inclusion criteria were broad as a mean to provide physical activity to as many aged adults as possible. Future studies should continue focusing on interventional programs that transition from experimental to real-world situations.

## 5. Conclusions

A multicomponent exercise program was found to be both safe and effective in improving functional fitness in community-dwelling aged adults and older adults. Low-cost community exercise programs seem to be a valuable and important resource for promoting physical activity and improving health in aged and older adults. These programs could provide a supportive and inclusive environment for individuals of advanced ages and fitness levels to engage in physical activity, improving their overall health and well-being. These findings suggest that an inclusive as well as cost-effective and community-based program can be deployed for all aged adults.

## Figures and Tables

**Table 1 medicina-59-00371-t001:** Results of study endpoints of the total sample (*n* = 48) at baseline and after exercise program.

Variables	Baseline				After		t	*p*	d
	M	SD	S	K	M	SD			
Chair stand test (repetitions)	21.04	5.80	0.49	−0.08	23.48	5.75	−14.06	<0.001	0.42
Arm curl (repetitions)	27.50	6.47	0.45	−0.32	31.52	7.41	−12.10	<0.001	0.58
Chair sit and reach (cm)	2.03	10.37	−1.38	1.54	2.58	9.85	−5.04	<0.001	0.05
Back scratch (cm)	−7.20	8.60	−0.04	−0.47	−5.74	7.81	−6.68	<0.001	0.18
Timed up and go (s)	4.64	0.82	0.97	0.88	4.48	0.85	5.60	<0.001	0.19
2 min step test (repetitions)	104.88	32.68	0.75	0.89	112.71	32.37	−9.41	<0.001	0.24
Hand grip strength (kg)	28.30	7.38	1.02	1.04	29.19	6.90	−3.33	0.002	0.15
Body Mass Index (kg/m^2^)	26.70	3.16	0.62	−0.06	26.55	3.19	5.16	<0.001	0.05

Notes: M = Mean; SD = Standard-Deviation; S = Skewness; K = Kurtosis; t, independent sample *t*-test; d = Cohen’s d; Δ(%) = changes between baseline and after exercise program.

**Table 2 medicina-59-00371-t002:** Descriptive statistics of each group at baseline and after exercise program.

Variables	<65 Years		≥65 Years		<65 Years		≥65 Years	
	Baseline		Baseline		After		After	
	M	SD	M	SD	M	SD	M	SD
Chair stand test	21.45	6.45	20.69	5.29	24.05	6.07	23.00	5.55
Arm curl	27.36	7.07	27.62	6.05	31.32	7.61	31.69	7.38
Chair sit and reach	2.77	9.79	1.40	11.00	3.23	9.40	2.04	10.37
Back scratch	−4.41	8.17	−9.56	8.38	−3.23	7.49	−7.87	7.58
Timed up and go	4.47	0.66	4.79	0.92	4.29	0.79	4.64	0.89
2 min step test	106.68	33.09	103.35	32.91	114.68	33.37	111.04	32.07
Hand grip strength	30.07	7.37	26.81	7.18	30.68	6.59	27.93	7.02
Body Mass Index	26.36	3.30	26.99	3.07	26.22	3.31	26.82	3.14

**Table 3 medicina-59-00371-t003:** Descriptive statistics of each group at baseline and after exercise program.

Variables	Mean Square	F	df1	df2	*p*	η_p_^2^	Pairwise Comparisons
Chair stand test							
Time	127.68	180.95	1	46	<0.001	0.90	before ≠ after
Group	24.05	0.28	1	46	0.60	0.01	not significant
Time*Group	0.73	0.91	1	46	0.35	0.04	not significant
Arm curl							
Time	348.01	101.17	1	46	<0.001	0.83	before ≠ after
Group	3.28	0.04	1	46	0.85	0.00	not significant
Time*Group	0.01	0.01	1	46	0.94	0.00	not significant
Chair sit and reach							
Time	6.01	28.93	1	46	<0.001	0.58	before ≠ after
Group	2.23	0.21	1	46	0.91	0.00	not significant
Time*Group	0.10	0.10	1	46	0.28	0.01	not significant
Back scratch							
Time	41.59	32.09	1	46	<0.001	0.60	before ≠ after
Group	366.14	3.70	1	46	0.07	0.15	not significant
Time*Group	0.82	0.82	1	46	0.38	0.04	not significant
Timed up and go							
Time	0.60	22.41	1	46	<0.001	0.52	before ≠ after
Group	2.23	1.62	1	46	0.22	0.07	not significant
Time*Group	0.01	0.48	1	46	0.49	0.02	not significant
2 min step test							
Time	1392.05	96.49	1	46	<0.001	0.82	before ≠ after
Group	30.73	0.01	1	46	0.92	0.00	not significant
Time*Group	0.05	0.00	1	46	0.96	0.00	not significant
Hand grip strength							
Time	15.67	20.04	1	46	<0.001	0.49	before ≠ after
Group	316.04	4.34	1	46	0.52	0.17	not significant
Time*Group	1.18	0.77	1	46	0.39	0.04	not significant
Body Mass Index							
Time	0.48	22.09	1	46	<0.001	0.51	before ≠ after
Group	9.41	0.47	1	46	0.50	0.02	not significant
Time*Group	0.01	0.16	1	46	0.70	0.01	not significant

Notes: F = f statistics; df1 and df2 = degrees of freedom; *p* = *p*-value; η_p_^2^ = partial eta square.

## Data Availability

Data are available upon request from the contact author.

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
