# Peer review of "Effects of a 24-Week Low-Cost Multicomponent Exercise Program on Health-Related Functional Fitness in the Community-Dwelling Aged and Older Adults"

_medicina, 2023, doi:10.3390/medicina59020371_

Round 1
Reviewer 1 Report
The overall aim of this study was to evaluate the effects of a multicomponent exercise program on the health-related functional fitness in community-dwelling aged and older adults. The authors placed an emphasis on the idea that this intervention was intended to be a long-term, low-cost community intervention. It is important that more studies such as this one be published. It is a strength to evaluate real world application of knowledge we have gained through laboratory studies. The authors stated it best at the end of the manuscript when they said, “The majority of exercise programs in older individuals have been conducted in laboratory settings, limiting their replicability in the community. In addition, inclusion criteria were broad, as a mean to provide physical activity to as much aged adults as possible. Future studies should continue focusing on interventional programs that transition from experimental to real-world situations.”
While this study is relevant and can add to the literature, there are several components of the article that can be addressed in order to improve the manuscript.
General
· The article is understandable but has many sentence structure and punctuation errors.
· There are several sentences with extra words. General editing needed.
Specific
· Abstract
o Line 21 – intention-to-treat analysis is mentioned here. This analysis is not mention at any other point in the article (methods/statistical analysis, results, or discussion).
· Introduction
o Lines 49-50 – “maintenance of flexibility generates force and power”
§ I question the validity of this statement. Please provide a reference for this statement as it is not found in reference #6 that is cited at the end of the sentence.
o Lines 65-66 – “this kind of approach is the most accessible training approach for these individuals.”
§ Please provide a reverence providing evidence that the described approach is most accessible or remove this sentence if a citation is not available.
o Line 67 – “in order to enable this…”
§ Please clarify what “this” is referring to.
o Lines 67-69
§ Consider rewording this sentence. It may not be accurate that “the elderly will require community programs… in order for them to live independently…”
§ It may be helpful, but not required.
o I appreciate lines 71-72 and 78-80. These are important statements in the manuscript.
o Line 91 – Please elaborate on the programs or type of programs you are referring to with “public health programs”
o Lines 93-94 – consider rephrasing this sentence.
§ “The literature has described…” is not grammatically correct because the literature did not describe something. Rather, something was described in the literature.
o Line 107 – Please describe “this sort of intervention”
§ It is unclear exactly what is meant here.
o Line 112 – “…they are adaptable…”
§ Is “they” referring to community exercise programs? Please clarify.
o Line 114 - …this method may assist…”
§ Please clarify what is meant by “this method”
o Line 127 – “The exercise intervention was anticipated to have a good and substantial effect…”
§ Consider replacing “good” with “positive”
§ This also applies to lines 176 and 349
· Methods
o Lines 157-158 – this is the second time the signing of an informed consent was mentioned (also mentioned in lines 142-143). Review the entire manuscript for redundancy. There are several statements that are mentioned twice.
o Line 165 – how did you define “physical inactivity”? Please define.
o Line 169 – states that “Participants needed to be available to participate in every… session” but line 171 clarifies that they only needed to participate in 75% of the sessions. Please fix this inconsistency.
o Lines 182-185 – It is unclear if the exercises are individualized of if each individual in the group is prescribed the same exercise (both type and intensity). Please clarify.
o Line 201 – It is implied that the 1-10 Borg Scale was used, not the 6-20 Borg Scale. Please specifically state the specific scale used.
o Lines 202-224 – In general, this paragraph lacks the detail necessary to replicate this portion of the protocol.
§ A couple of the questions that came to mind that will help and specificity are:
· How many resistance training lifts/motions were performed each session?
· How did the trainer decide which lifts to incorporate each session?
· Did the number of lifts decrease as sets increased in order to stay within the designated time?
· Reps were defined with a range 12-15. How did each participant know when to stop within that range?
o Line 217 – “…distance between participants was determined;”
§ How was distance determined? Based on skill level?
o Lines 274-280 – for replicability, please define the position of the participant during the handgrip test (i.e. sitting or standing, arm angle, etc.)
o Statistical analysis – no mention of intention-to-treat. It appears to me that this analysis was not needed because all participants fulfilled the requirement to attend 75% of the intervention sessions.
· Results
o Line 309 – number of males and females not reported.
§ It would also be beneficial if number of males and females were defined for each group represented in Table 2.
o Consider revising the titles of you table to not include “descriptive”
§ Descriptive should include sex, height, weight, age, etc.
o Consider adding height and weight to the results. The reason is related to a comment in the discussion section.
· Discussion
o Line 387 – Please consider rewording this sentence. I think there is a typo in the sentence that makes it hard to understand.
o Lines 396-405 – Please reconsider this paragraph. You are discussing changes in body composition yet, you report no outcomes that allow you to make statements regarding body composition. BMI does not tell us if body composition has changed. The only thing you can say has changed is weight (this is why I suggest reporting weight in the results section). Even if you rewrite this paragraph and solely address BMI, do it with caution because the change in BMI/weight is so small that I would argue it has no clinical or health related significance (despite having statistical significance).
o Line 431 – It is state that high adherence was due to the exercise physiologist providing motivation. This statement cannot be substantiated without qualitative data from the participants. If this data does not exist, please refrain from making the inference. If the data does exist, please report the data.
o Limitations – one of the limitations of this study that was not mentioned is that there was no mid-intervention data collection. Tracking progress through the training period could have added strength to this study.
§ Otherwise, the strengths and limitations are well reported.
Author Response
Reviewer 1
The overall aim of this study was to evaluate the effects of a multicomponent exercise program on the health-related functional fitness in community-dwelling aged and older adults. The authors placed an emphasis on the idea that this intervention was intended to be a long-term, low-cost community intervention. It is important that more studies such as this one be published. It is a strength to evaluate real world application of knowledge we have gained through laboratory studies. The authors stated it best at the end of the manuscript when they said, “The majority of exercise programs in older individuals have been conducted in laboratory settings, limiting their replicability in the community. In addition, inclusion criteria were broad, as a mean to provide physical activity to as much aged adults as possible. Future studies should continue focusing on interventional programs that transition from experimental to real-world situations.”
Response: We appreciate your positive comments.
While this study is relevant and can add to the literature, there are several components of the article that can be addressed in order to improve the manuscript.
Response: Point-by-point responses as provided and revisions are tracked in the manuscript using the track change option in MS Word.
General
- The article is understandable but has many sentence structure and punctuation errors.
Response: The entire manuscript was revised.
- There are several sentences with extra words. General editing needed.
Response: The entire manuscript was revised.
Specific
- Abstract
o Line 21 – intention-to-treat analysis is mentioned here. This analysis is not mentioned at any other point in the article (methods/statistical analysis, results, or discussion).
Response: We removed this reference as the current study was quasi-experimental in nature and not randomized.
- Introduction
o Lines 49-50 – “maintenance of flexibility generates force and power”
- I question the validity of this statement. Please provide a reference for this statement as it is not found in reference #6 that is cited at the end of the sentence.
Response: Sentence was revised “The maintenance of flexibility, force and power production, and coordination is dependent on physical activity levels during previous stages in life, such as adulthood.”
o Lines 65-66 – “this kind of approach is the most accessible training approach for these individuals.”
- Please provide a reverence providing evidence that the described approach is most accessible or remove this sentence if a citation is not available.
Response: Sentence was removed as the empirical view of the authors is not currently support by existing literature.
o Line 67 – “in order to enable this…”
- Please clarify what “this” is referring to.
Response: Sentence was revised “Community programs that provide exercise interventions may be helpful aiming to provide the elderly to live independently, with enough physical and functional fitness.”
o Lines 67-69
- Consider rewording this sentence. It may not be accurate that “the elderly will require community programs… in order for them to live independently…”
- It may be helpful, but not required.
Response: Sentence was revised “Community programs that provide exercise interventions may be helpful aiming to provide the elderly to live independently, with enough physical and functional fitness.”
o I appreciate lines 71-72 and 78-80. These are important statements in the manuscript.
Response: We appreciate your positive comments.
o Line 91 – Please elaborate on the programs or type of programs you are referring to with “public health programs”
Response: Sentence was revised “This type of exercise training combines different types of exercise (e.g., strength, balance, gait, aerobic, and flexibility) in the same exercise session, and it is a public health program option aiming to maintain or even improve physical and cognitive functions of community-dwelling aged and older adults.”
o Lines 93-94 – consider rephrasing this sentence.
- “The literature has described…” is not grammatically correct because the literature did not describe something. Rather, something was described in the literature.
Response: Sentence was revised “Researchers have described the unique effects of an acute multicomponent exercise program to slow functional decline in older individuals.”
o Line 107 – Please describe “this sort of intervention”
- It is unclear exactly what is meant here.
Response: Sentence was revised “To the best of our knowledge, multicomponent exercise programs has been utilized in the community adults over the age of 65, however protocols are not always described in detail for replication” and complemented afterwards.
o Line 112 – “…they are adaptable…”
- Is “they” referring to community exercise programs? Please clarify.
Response: Yes, sentence was revised “It is critical that community programs would involve people of all ages (remembering that physical demands is not limited to those over 65), and that these community exercise programs are adaptable and effective for everyone.”
o Line 114 - …this method may assist…”
- Please clarify what is meant by “this method”
Response: Sentence was revised “Because exercise improves long-term quality of life [22], this type of physical activity may assist stakeholders in providing efficient interventions to delay or perhaps pre-vent age-related functional decline in their communities, hence increasing health-related quality of life in community-living older individuals.”
o Line 127 – “The exercise intervention was anticipated to have a good and substantial effect…”
- Consider replacing “good” with “positive”
- This also applies to lines 176 and 349
Response: We appreciate your review. The word “good” was replaced by “positive” in all mentioned lines.
- Methods
o Lines 157-158 – this is the second time the signing of an informed consent was mentioned (also mentioned in lines 142-143). Review the entire manuscript for redundancy. There are several statements that are mentioned twice.
Response: Sentence was removed. We also tracked for other redundancies.
o Line 165 – how did you define “physical inactivity”? Please define.
Response: Inclusion criteria was revised “being not engaged in any type of exercise program”
o Line 169 – states that “Participants needed to be available to participate in every… session” but line 171 clarifies that they only needed to participate in 75% of the sessions. Please fix this inconsistency.
Response: Indication to participate voluntarily in a 3-times-a-week session of physical exercise program to each individual was made prior to conducting the exercise protocol. However, if at the end of the program they did not comply to participate in at least 75% of all sessions, data from them would not be included for analysis. This procedure has been adopted in other referenced studies (Carvalho et al., 2009; García-Molina et al., 2018)
o Lines 182-185 – It is unclear if the exercises are individualized of if each individual in the group is prescribed the same exercise (both type and intensity). Please clarify.
Response: Sentence was revised “Each individual in the group performed the same exercises and adaptations were made when necessary.”
o Line 201 – It is implied that the 1-10 Borg Scale was used, not the 6-20 Borg Scale. Please specifically state the specific scale used.
Response: Reference to the 10-point Borg Perceived Exertion Scale was added.
o Lines 202-224 – In general, this paragraph lacks the detail necessary to replicate this portion of the protocol.
Response: The entire intervention section was revised. We believe that the protocol is described in meticulous details now, worthy of replication for future studies.
- A couple of the questions that came to mind that will help and specificity are:
- How many resistance training lifts/motions were performed each session?
- How did the trainer decide which lifts to incorporate each session?
Response: See previous comment and the intervention section for details.
- Did the number of lifts decrease as sets increased in order to stay within the designated time?
Response: As described in the manuscript “Resistance exercise lasted between 15-30 minutes”. Hence, we accounted for the increase in volume and time.
- Reps were defined with a range 12-15. How did each participant know when to stop within that range?
Response: All participants had the ability to count. If the participant would be able to perform 12 repetitions with light-moderate effort, the exercise physiologist would indicate to perform 3 more repetitions. This information was added to the manuscript.
o Line 217 – “…distance between participants was determined;”
- How was distance determined? Based on skill level?
Response: “Distance” refers to the distance between participants to execute safely balance exercises.
o Lines 274-280 – for replicability, please define the position of the participant during the handgrip test (i.e., sitting or standing, arm angle, etc.)
Response: Information was added “The position for assessing hand grip strength was in a sitting position, with the shoulder in neutral position, the elbow extended at 180°, and the forearm and wrist in neutral position.”
o Statistical analysis – no mention of intention-to-treat. It appears to me that this analysis was not needed because all participants fulfilled the requirement to attend 75% of the intervention sessions.
Response: We agree with the reviewer. We removed this reference as the current study was quasi-experimental in nature and not randomized.
- Results
o Line 309 – number of males and females not reported.
- It would also be beneficial if number of males and females were defined for each group represented in Table 2.
Response: Number of male and female participants were added.
o Consider revising the titles of you table to not include “descriptive”
- Descriptive should include sex, height, weight, age, etc.
Response: Title of the table was revised to “Table 1. Results of study endpoints of the total sample (n = 48) at baseline and after exercise program.”
o Consider adding height and weight to the results. The reason is related to a comment in the discussion section.
Response: We believe that adding the data from height before and after the program would no contribute significantly to the study objective. In fact, there is evidence supporting that height could increase or decrease due to physical exercise in aged and older adults.
- Discussion
o Line 387 – Please consider rewording this sentence. I think there is a typo in the sentence that makes it hard to understand.
Response: The paragraph was revised “Participants increased their flexibility since lower and upper limb tests revealed significant improvements (p<0.001). While there are some inconsistencies regarding the benefits of flexibility training [43], current results support previous research examining the effects of exercise on flexibility [9,41], showing that a multicomponent exercise program can increase flexibility in aged and older adults. The significant improvements could also be related to the exercises conducted in the strength component, as range of motion is increased by this sort of training [9].”
o Lines 396-405 – Please reconsider this paragraph. You are discussing changes in body composition yet; you report no outcomes that allow you to make statements regarding body composition. BMI does not tell us if body composition has changed. The only thing you can say has changed is weight (this is why I suggest reporting weight in the results section). Even if you rewrite this paragraph and solely address BMI, do it with caution because the change in BMI/weight is so small that I would argue it has no clinical or health related significance (despite having statistical significance).
Response: We partially agree with the reviewer. BMI is an indicator of body composition, specifically in the older adult population and thus should not be discharged from discussion. However, we do agree that the effect is small. Thus, we revised this paragraph.
o Line 431 – It is state that high adherence was due to the exercise physiologist providing motivation. This statement cannot be substantiated without qualitative data from the participants. If this data does not exist, please refrain from making the inference. If the data does exist, please report the data.
Response: Paragraph was removed since we have no data to support our claim.
o Limitations – one of the limitations of this study that was not mentioned is that there was no mid-intervention data collection. Tracking progress through the training period could have added strength to this study.
Response: Limitation added.
- Otherwise, the strengths and limitations are well reported.
Response: We appreciate your positive feedback.
Reviewer 2 Report
This study aimed to examine the effects of a multi-component exercise program on health-related functional fitness in community-dwelling aged and older adults. Based on this objective, the authors conclude that A multi-component exercise program was safe and effective in improving functional fitness in community-dwelling aged and older adults. Also, the authors state that the training protocol applied was of interest to both aged and older adults since adherence to the 24-week program was greater than 75%. Thus, this article tries to show that a multi-component program is feasible for older adults. However, several points can be improved and clarified in the manuscript. Below, I pointed out some aspects that can be improved.
Main points:
- I do not see the importance of comparing older people based on age since we have articles studying octogenarians and nonagenarians. Besides, most studies with older people use 60 or 65 years old as an inclusion criterion. However, it is possible to explore the argument about low-cost interventions.
- No novelty in the study since there are already several papers using multi-component training and applying the senior fitness test battery or others types of functional fitness evaluation.
- Also, there is no rationality in the introduction to justify the possible differences in training-related adaptations between different ages in older people.
- Regarding sample size calculation, the effect size used was based on what? It is essential to put this information in the document to understand your recruitment.
- The information about the differences in age between groups needs to be exposed, which is essential to understand the results. Also, the mean and standard deviation of the age of the participants do not seem to show significant differences to allow the aimed comparison.
- How many women and men composed the sample?
- Are you basing the affirmations about clinically meaningful improvements on which criteria? This needs to be explained.
- The results do not allow you to talk about different types of older people since it is not exposed the means and standard deviation in each group analyzed. Also, there were only trivial, small, and moderate effect sizes. Thus, it is precipitated to talk about good effects.
- The explanation for the improvements in the strength should be more profound and explore the effect sizes found to give the reader a valuable interpretation of the results compared to the literature.
- If strength training is enough to improve flexibility, why have a flexibility part in the training session?
- The benefits found in cardiorespiratory fitness could be related to the circuit training applied instead of the endurance activity. The training protocol does not allow the statement that the cardiorespiratory adaptations are related to endurance activities.
- The results do not allow speculation about body composition changes since only BMI was evaluated, and the effect size was trivial. I suggest a different approach to discussing this topic.
- Several articles investigated long-term interventions, so I recommend being more cautious with the statements in the discussion.
- The adherence does not state that the training protocol is interesting. After the study, a questionnaire about satisfaction or motivation to keep practicing should be used to investigate the participants’ interest in the intervention.
- Supposing no comparison was made between resistance or endurance training with multi-component training. It would be best if you did not affirm that multi-component training could replace the abovementioned options since both efficiently promote health.
Minor corrections:
- The introduction gives much attention to low-cost interventions in the context of aging. So, I suggest changing the title to “Effects of a 24-week low-cost multi-component physical training on health-related functional fitness in the community-dwelling aged and older adults”. Also, I recommend exposing the costs involved in developing the intervention, including the equipment and the service of the professionals responsible for the supervision.
- It is exposed that “data suggests that minimum intervention lengths of 6 months may be required to elicit changes in aged people”. However, many articles show improvements using multi-component or functional training in a short period (i.e., less than eight weeks). So, I recommend rewriting this sentence.
- Along the same line, the intervention’s duration and sample size are also highlighted as gaps in the literature. However, some articles used more than 24 weeks and with greater sample sizes than the number used in this study.
- I wonder if this type of intervention is underutilized since the most recent guidelines for exercise prescription for older people recommend a multi-component approach, and both multi-component and functional training are often used among older people.
- I believe that handgrip is more used as an indicator of general health and associated with mortality than necessarily a quality-life-related measure.
- It is exposed that was found a very good intra-observer validity. However, Cohen’s values between 0.60 and 0.80 are just substantial. Besides, for intra-observer reliability, the most appropriate analysis is the intraclass correlation. Thus, I recommend adjusting this point.
- Was there just one professional to supervise the training sessions? This needs to be clearly explained in the manuscript.
- Was there some form of anchoring the effort perception to ensure the participants knew the maximum effort? This should be exposed. Also, was it taken care of regarding the possible influence of the answer of one participant over the response of another participant?
- Why was the endurance training performed before the resistance training? Strength is an essential capacity to maintain the autonomy of older people, and circuit training is efficient in improving cardiorespiratory fitness. Finally, the interference effect is reduced when you perform resistance training before endurance training.
- The flexibility protocol needs to be better explained. What do you mean by “active: passive”? How was this protocol performed?
- The Mauchly sphericity test is applied when you have three or more measurements. Thus, it does not apply to this study.
- You use percentual changes in the tables. However, the percentual values can underestimate high absolute values and overestimate low values. Thus, I suggest removing this information and paying more attention to the effect sizes.
- You should put Cohen’s d in table 2 since you analyzed the time effect and the p values for each test.
- The information presented in Table 3 might be exposed in the text.
- The reference list must be standardized. There are some discrepancies in the names of the journals.
Based on these comments, I hope you can improve the quality of the manuscript.
Author Response
Reviewer 2
This study aimed to examine the effects of a multi-component exercise program on health-related functional fitness in community-dwelling aged and older adults. Based on this objective, the authors conclude that A multi-component exercise program was safe and effective in improving functional fitness in community-dwelling aged and older adults. Also, the authors state that the training protocol applied was of interest to both aged and older adults since adherence to the 24-week program was greater than 75%. Thus, this article tries to show that a multi-component program is feasible for older adults. However, several points can be improved and clarified in the manuscript. Below, I pointed out some aspects that can be improved.
Response: Point-by-point responses as provided and revisions are tracked in the manuscript using the track change option in MS Word.
Main points:
- I do not see the importance of comparing older people based on age since we have articles studying octogenarians and nonagenarians. Besides, most studies with older people use 60 or 65 years old as an inclusion criterion. However, it is possible to explore the argument about low-cost interventions.
Response: We like to point out that we are not comparing octogenarians and nonagenarians. This study had two objectives: a) to examine the effects of a low-cost multicomponent exercise program on health-related functional fitness in the community-dwelling aged and older adults. b) to compare the exercise program between aged adults (<65 years) and those considered elderly (≥65 years). Please see our response to your following comment as a complement to our response in this one.
- No novelty in the study since there are already several papers using multi-component training and applying the senior fitness test battery or other types of functional fitness evaluation.
Response: Novelty is described in lines 105-123.
- Also, there is no rationality in the introduction to justify the possible differences in training-related adaptations between different ages in older people.
Response: We argue the need for this study in lines 105-123.
- Regarding sample size calculation, the effect size used was based on what? It is essential to put this information in the document to understand your recruitment.
Response: We clarified the selected effect size in the manuscript “. The calculation of the effect was based on studies using similar protocols and de-signs converting reported Cohen’s d effect sized into f.”
- The information about the differences in age between groups needs to be exposed, which is essential to understand the results. Also, the mean and standard deviation of the age of the participants do not seem to show significant differences to allow the aimed comparison.
Response: As mentioned earlier, a paragraph was inserted.
- How many women and men composed the sample?
Response: Number of male and female participants were added.
- Are you basing the affirmations about clinically meaningful improvements on which criteria? This needs to be explained.
Response: Sentence was revised “Significant improvements were observed in the…”
- The results do not allow you to talk about different types of older people since it is not exposed the means and standard deviation in each group analyzed. Also, there were only trivial, small, and moderate effect sizes. Thus, it is precipitated to talk about good effects.
Response: Mean and standard deviation data for each group is now reported.
- The explanation for the improvements in the strength should be more profound and explore the effect sizes found to give the reader a valuable interpretation of the results compared to the literature.
Response: The paragraph was revised.
- If strength training is enough to improve flexibility, why have a flexibility part in the training session?
Response: The paragraph was revised.
- The benefits found in cardiorespiratory fitness could be related to the circuit training applied instead of the endurance activity. The training protocol does not allow the statement that the cardiorespiratory adaptations are related to endurance activities.
Response: The paragraph was revised.
- The results do not allow speculation about body composition changes since only BMI was evaluated, and the effect size was trivial. I suggest a different approach to discussing this topic.
Response: The paragraph was revised.
- Several articles investigated long-term interventions, so I recommend being more cautious with the statements in the discussion.
Response: We agree and thus sentence was revised.
- The adherence does not state that the training protocol is interesting. After the study, a questionnaire about satisfaction or motivation to keep practicing should be used to investigate the participants’ interest in the intervention.
Response: Paragraph was removed since we have no data to support our claim.
- Supposing no comparison was made between resistance or endurance training with multi-component training. It would be best if you did not affirm that multi-component training could replace the abovementioned options since both efficiently promote health.
Response: The entire conclusion section was revised.
Minor corrections:
- The introduction gives much attention to low-cost interventions in the context of aging. So, I suggest changing the title to “Effects of a 24-week low-cost multi-component physical training on health-related functional fitness in the community-dwelling aged and older adults”.
Response: Title was revised.
Also, I recommend exposing the costs involved in developing the intervention, including the equipment and the service of the professionals responsible for the supervision.
Response: Due to institutional regulations, we are unable to disclosure costs that this protocol had on recruiting two exercise physiologists. The costs related to the equipment would bias readers as the equipment cost may vary according to country. As the name implies, and equipment described in the method section, we used low-cost material that are portable, easy to pack up, and accessible to acquire.
- It is exposed that “data suggests that minimum intervention lengths of 6 months may be required to elicit changes in aged people”. However, many articles show improvements using multi-component or functional training in a short period (i.e., less than eight weeks). So, I recommend rewriting this sentence.
Response: Sentence was removed and we took a different perspective of the novelty of this study.
- Along the same line, the intervention’s duration and sample size are also highlighted as gaps in the literature. However, some articles used more than 24 weeks and with greater sample sizes than the number used in this study.
Response: We agree with the reviewer and discuss gaps and limitations of existing studies differently.
- I wonder if this type of intervention is underutilized since the most recent guidelines for exercise prescription for older people recommend a multi-component approach, and both multi-component and functional training are often used among older people.
Response: The entire paragraph was revised.
- I believe that handgrip is more used as an indicator of general health and associated with mortality than necessarily a quality-life-related measure.
Response: As described in the manuscript “Low scores on this measure are linked to several negative health outcomes, including chronic morbidities, functional impairments, and all-cause death. On the contrary, high scores on hand grips strength are associated with greater mental health, overall strength, autonomy, and longevity.” Thus, the agree with your statement in full.
- It is exposed that was found a very good intra-observer validity. However, Cohen’s values between 0.60 and 0.80 are just substantial. Besides, for intra-observer reliability, the most appropriate analysis is the intraclass correlation. Thus, I recommend adjusting this point.
Response: We appreciate your concern. ICC value was calculated and reported in the manuscript.
- Was there just one professional to supervise the training sessions? This needs to be clearly explained in the manuscript.
Response: Two exercise professionals as described in line 152 and line 188. We clarified in the entire manuscript.
- Was there some form of anchoring the effort perception to ensure the participants knew the maximum effort? This should be exposed. Also, was it taken care of regarding the possible influence of the answer of one participant over the response of another participant?
Response: We exposed anchoring procedures “Looking at the Borg Perceived Exertion Scale, for each item we provided statements for anchoring perceived effort. For example, for scoring 1 we described “hardly any exertion” and for scoring 10 we described “feels almost impossible to keep going”.
- Why was the endurance training performed before the resistance training? Strength is an essential capacity to maintain the autonomy of older people, and circuit training is efficient in improving cardiorespiratory fitness. Finally, the interference effect is reduced when you perform resistance training before endurance training.
Response: We could not find any research that confirms your statement, specifically considering older adults. The option of conducting cardiorespiratory training before resistance training had no specific justification except that during cardiorespiratory training, the exercise physiologist had the capacity to prepare the resistance training component avoiding waste of time.
- The flexibility protocol needs to be better explained. What do you mean by “active: passive”? How was this protocol performed?
Response: We changed the word passive for relaxed. 1:1 ration means that the time of stretched muscle was the same for the time of relaxed muscle.
- The Mauchly sphericity test is applied when you have three or more measurements. Thus, it does not apply to this study.
Response: We agree and made revisions. The Levene’s test was considered for homoscedasticity analysis.
- You use percentual changes in the tables. However, the percentual values can underestimate high absolute values and overestimate low values. Thus, I suggest removing this information and paying more attention to the effect sizes.
Response: Changes were removed.
- You should put Cohen’s d in table 2 since you analyzed the time effect and the p values for each test.
Response: Table 2 is just for descriptive purpose as it complemented by Table 3.
- The information presented in Table 3 might be exposed in the text.
Response: We believe that the data reported in Table 3 should be maintained in the Table for reading clarity and transparency.
- The reference list must be standardized. There are some discrepancies in the names of the journals.
Response: The entire reference list was revised.
Based on these comments, I hope you can improve the quality of the manuscript.
Response: We appreciate your comments and hope that our responses and revisions are in accordance with the objective in improving the manuscript.
Round 2
Reviewer 1 Report
Thank you for addressing all my comments!
Reviewer 2 Report
I appreciate the authors' efforts to improve the manuscript and attend to previous recommendations. There are some significant improvements compared to the first version, so congratulations to the authors. However, a few points need to be addressed, which are listed below.
Major points
- I recommend combining tables 2 and 3 since they are related to the same comparison. To attend to this recommendation, insert the f values and degrees of freedom in the text and put the partial eta square and p values in table 2. You can also insert Cohen's d for time comparisons;
- The discussion about body composition is an equivocal point since the body mass index does not provide information about body fat and lean mass. So I suggest keeping the discussion on only the topics that were measured.
Minor corrections
- The abstract reported the effect size value for the back scratch but not for the other tests. So it is recommended to show the values for all tests in the abstract;
- I recommend inserting the ICC value for each test instead of a summary. Also, with the ICC values, it is helpful to show the standard error of measurement and minimum detectable difference since there is no control group in this study;
- Showing p-values in the discussion session is unnecessary, so I recommend removing them;
- One more time, the term "clinically significant" is used, but it is necessary to explain the criteria to assume that one change is clinically significant.
- There are yet to be standardized references related to journal name abbreviations. It is recommended to standardize the use of dots after the abbreviations.
Addressing the points mentioned above will improve the quality of the manuscript.
Author Response
Reviewer 2
The I appreciate the authors' efforts to improve the manuscript and attend to previous recommendations. There are some significant improvements compared to the first version, so congratulations to the authors. However, a few points need to be addressed, which are listed below.
Response: We appreciate your positive comments.
Major points
I recommend combining tables 2 and 3 since they are related to the same comparison. To attend to this recommendation, insert the f values and degrees of freedom in the text and put the partial eta square and p values in table 2. You can also insert Cohen's d for time comparisons;
Response: We appreciate your suggestion. For reading purpose we intend to maintain Table 2 and Table 3 separately. Again, Table 2 is not for comparison but rather for descriptive purpose. Writing f values and df in text would mar reading quality as the text would be confusing.
The discussion about body composition is an equivocal point since the body mass index does not provide information about body fat and lean mass. So, I suggest keeping the discussion on only the topics that were measured.
Response: We agree with the reviewer. Paragraph removed.
Minor corrections
The abstract reported the effect size value for the back scratch but not for the other tests. So, it is recommended to show the values for all tests in the abstract;
Response: Done.
I recommend inserting the ICC value for each test instead of a summary. Also, with the ICC values, it is helpful to show the standard error of measurement and minimum detectable difference since there is no control group in this study;
Response: It is not a standard procedure to report ICC values for each test. Considering cited research, none of the report ICC values. Thus, our report on the overall reliability analysis is acceptable.
Showing p-values in the discussion session is unnecessary, so I recommend removing them;
Response: Removed.
One more time, the term "clinically significant" is used, but it is necessary to explain the criteria to assume that one change is clinically significant.
Response: Term was revised for “substantial”.
There are yet to be standardized references related to journal name abbreviations. It is recommended to standardize the use of dots after the abbreviations.
Response: References revised.
Addressing the points mentioned above will improve the quality of the manuscript.
Response: We appreciate your positive comments.